# A Sandbox Tool to Bias(Stress)-Test Fairness Algorithms

## Abstract

Motivated by the growing importance of reducing unfairness in ML predictions, Fair-ML researchers have presented an extensive suite of algorithmic "fairness-enhancing" remedies. Most existing algorithms, however, are agnostic to the sources of the observed unfairness. As a result, the literature currently lacks guiding frameworks to specify conditions under which each algorithmic intervention can potentially alleviate the underpinning cause of unfairness. To close this gap, we scrutinize the underlying biases (e.g., in the training data or design choices) that cause observational unfairness. We present the conceptual idea and a first implementation of a bias-injection sandbox tool to investigate fairness consequences of various biases and assess the effectiveness of algorithmic remedies in the presence of specific types of bias. We call this process the bias(stress)-testing of algorithmic interventions. Unlike existing toolkits, ours provides a controlled environment to counterfactually inject biases in the ML pipeline. This stylized setup offers the distinct capability of testing fairness interventions beyond observational data and against an unbiased benchmark. In particular, we can test whether a given remedy can alleviate the injected bias by comparing the predictions resulting after the intervention in the biased setting with true labels in the unbiased regime – that is, before any bias injection. We illustrate the utility of our toolkit via a proof-of-concept case study on synthetic data. Our empirical analysis showcases the type of insights that can be obtained through our simulations.

## 1 Introduction

Machine Learning (ML) increasingly makes or informs high-stakes decisions allocating or withholding vital resources to individuals and communities in domains such as employment, credit lending, education, welfare benefits, and beyond. If not done carefully, ML-based decision-making systems may worsen existing inequities and impose disparate harms on already underserved individuals and social groups. This realization has motivated an active area of research into quantifying and guaranteeing fairness for ML. Prior work has proposed various mathematical formulations of (un)fairness as predictive (dis)parities (Berk et al., 2018; Dwork et al., 2012; Hardt et al., 2016; Joseph et al., 2016) and fairness-enhancing algorithms to guarantee the respective parity conditions in the trained model's predictions (Agarwal et al., 2018; Hardt et al., 2016; Calmon et al., 2017; Feldman et al., 2015; Zhang et al., 2018; Kamiran et al., 2012). Our work argues that these interventions are not sufficiently well-understood to warrant practical uptake. One crucial limitations of these algorithms is the fact that they are agnostic to the underlying *sources* of the observed unfairness. As a result, applying them in practice may simply hide the real problem by ensuring narrowly defined notions of parity in predictions. As a result, what these methods seemingly gain in observational parity can come at the cost of predictive disparity and accuracy loss in deployment, and at worst, they can become an instrument of fair-washing (Aïvodji et al., 2019).

As an example, consider a hypothetical healthcare setting in which electronic healthcare data is used to determine which patients are selected for a specialized treatment. Assume the hospital wants to ensure their prediction model is fair across a demographic majority and minority group. In this setting, it may seem intuitive to promote fairness by enforcing an Equalized Odds constraint. Equalized Odds ensures that, given the true need for the procedure is the same, the model's decision to select a patient is independent of the patient's group membership. While this may seem like a viable solution to combat unfairness, it is agnostic

to the types of data bias that cause outcome disparities. Training data in healthcare prediction tasks like this is often plagued by biases (Obermeyer et al., 2019; Chen et al., 2021). In our example, we may not have access to patients' true need for the procedure and instead default to healthcare cost as a proxy outcome to train the model. Since access to healthcare has historically been lower for some minority groups, this can lead to a setting in which minority group patients selected for the procedure are sicker than their majority group counterparts even when enforcing Equalized Odds. Blindly applying an off-the-shelf Equalized Odds fairness enhancing method without understanding the types of bias present in this setting could thus hide the real problem while creating an illusion of fairness.

Our work aims to address the above shortcoming by offering a simulation framework for examining fairness interventions in the presence of various biases. This offers an initial yet crucial step toward a broader research agenda: to trace the limitations and scope of applicability of fairness-enhancing algorithms. We start with the observation that the ML pipeline consists of numerous steps, and distinct types of biases (e.g., under-/over-representation of certain groups, label bias, or measurement bias in the training data) can creep into it at various stages, amplifying or concealing each other in the trained model's predictive disparities. The fair-ML scholarship currently lacks a comprehensive framework for specifying the conditions under which each algorithmic fairness-enhancing mechanism effectively removes specific types of biases—instead of simply covering up their manifestations as unfairness. For example, it is unclear what type of intervention (e.g., pre-, in-, or post-processing) one must employ depending on the underlying cause of the observed statistical disparity. As a concrete instance, a careful investigation of the relationship between biases and fairness remedies may reveal that if the source of unfairness is label bias among examples belonging to the disadvantaged group, imposing fairness constraints on ERM may be more effective than certain types of pre-processing or post-processing techniques. The reverse might be true for a different bias (e.g., biased choice of hypothesis class).

**Our simulation tool**. Motivated by the above account, in this work, we identify and simulate various (stylized) forms of bias that can infiltrate the ML pipeline and lead to observational unfairness. We prototype a sandbox toolkit designed to facilitate simulating and assessing the effectiveness of algorithmic fairness methods in alleviating specific types of bias, by providing a controlled environment. We call this process the *bias(stress)-testing* of algorithmic interventions. Our sandbox offers users a simulation environment to stress-test existing remedies by

1. simulating/injecting various types of biases (e.g., representation bias, measurement bias, omitted variable bias, model validity discrepancies) into their ML pipeline;

2. observing the interactions of these biases with one another via the predictions produced at the end of the ML pipeline (i.e., through the trained model);

3. and testing the effectiveness of a given algorithmic fairness intervention in alleviating the injected biases.

This paper offers a preliminary implementation of the idea (see footnote 1) along with a detailed proof-of-concept analysis showing its utility. The sandbox is currently realized as a python library and we are working to add a visual user interface component in the future. We emphasize that the tool needs to be further developed and thoroughly evaluated before it is ready to be utilized beyond educational and research settings. The current implementation can be utilized

- in research settings to explore the relationships between bias and unfairness, and shape informed hypotheses for further theoretical and empirical investigations;

- as an educational tool to demonstrate the nuanced sources of unfairness, and grasp the limitations of fairness-enhancing algorithms;

Ultimately once the tool is fully developed and validated, we hope that it can be utilized by practitioners interested in exploring the potential effect of various algorithmic interventions in their real-world use cases.

This will be an appropriate usage of the tool *if* (and this is an crucial if) the bias patterns in the real-world data are well-understood.

**Counterfactual comparisons**. The key idea that distinguishes our tool from existing ones is the possibility of evaluating fairness interventions beyond observational measures of predictive disparity. In particular, we can test whether a given remedy can alleviate the injected bias by comparing the predictions resulting from the intervention in the biased setting with the true labels *before* bias injection. This ability to compare with the unbiased data provides an ideal baseline for assessing the efficacy of a given remedy. We note, however, that the viability of this approach requires access to unbiased data. We, therefore, strongly recommend restricting the use of our tool to *synthetic* data sets—unless the user has a comprehensive and in-depth understanding of various biases in the real-world dataset they plan to experiment with.

**Remark 1** *Note that in the case of real-world applications, one can rarely assume the training data are free of bias. However, if the practitioner is aware of what biases are present in the data (e.g., under-representation of a specific group) our toolkit may still allow them to obtain practically relevant insights concerning the effect of their fairness interventions of choice (e.g., up-sampling) on alleviating that bias—assuming* that we can *extrapolate the observed relationship between the amount of* additional *bias injected and the trained model's unfairness. We leave a thorough assessment of our toolkit's applicability to real-world data as a critical direction for future work.*

**Case study**. We demonstrate the utility of our proposed simulation tool through a case study. In particular, we simulate the setting studied in (Blum & Stangl, 2020). Blum and Stangl offer one of the few rigorous analyses of fairness algorithms under specific bias conditions. Their work establishes an intriguing theoretical result that calls into question conventional wisdom about the existence of tradeoffs between accuracy and fairness. In particular, their theoretical analysis shows that when underrepresentation bias is present in the training data, constraining Empirical Risk Minimization (ERM) with Equalized Odds (EO) conditions can recover a Bayes optimal classifier under certain conditions. Utilizing our tool, we investigate the extent to which these findings remain valid in the finite data case. Our findings suggest that, even for relatively simple regression models, a considerable amount of training data is required to recover from under-representation bias. In the studied settings with smaller data sets and little to moderate under-representation bias the intervention model showed to be no more successful in recovering the Bayes optimal model than a model without intervention. We then investigate the effectiveness of the in-processing EO intervention when alternative types of biases are injected into the training data. We observe that the intervention model struggles to recover from the other studied types of biases. In some of the bias settings, such as a difference in base rates or differential label noise, the model with Equalized Odds intervention can provably not recover the Bayes optimal classifier since the latter does not fulfill Equalized Odds. Finally, we contrast the in-processing approach with the original post-processing method introduced by (Hardt et al., 2016) to ensure EO. As we discuss in Sections 3 and 4, our empirical analysis identifies several critical limitations of these methods.

**Remark 2** *Our proof-of-concept demonstration deliberately addresses a small number of fairness-enhancing algorithms, and evaluates them in the presence of a wide range of data biases. While the sandbox can be used to contrast a wide array of fairness definitions and algorithms, such an analysis is beyond the scope of the current contribution and is left as an important avenue for future work.*

In summary, we present a first implementation of the unified toolkit to bias(stress)-test existing fairness algorithms by assessing their performance under carefully injected biases. As demonstrated by our case study, our sandbox environment can offer practically relevant insights to users and present researchers with hypotheses for further investigations. Moreover, our work provides a potential hands-on educational tool to learn about the relationship between data bias and unfairness as a part of the AI ethics curricula. Once evaluated and validated carefully, we hope that this tool contributes to educating current and future AI experts about the potential of technological work for producing or amplifying social disparities, and in the process, impacting lives and society.

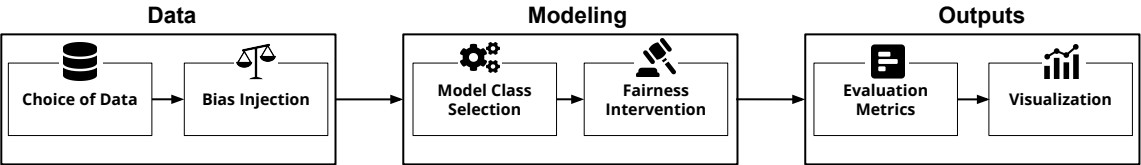

Figure 1: Flowchart illustrating the modules of the sandbox framework.

## 2 Description of the Sandbox

The proposed sandbox presents a tool to understand the effectiveness of fairness enhancing algorithms under counterfactually injected bias in binary classification settings. The tool can be visualized as a simplified ML pipeline, with room for customization at each step. As indicated by its name, the sandbox prioritizes modularity and allows users to play around and experiment with alternatives at every stage. We summarize the six stages of the pipeline, which are illustrated in Figure 1, in the following. Note that, at the current time, some of implementation details as well as a visual user interface are still under development.

1. **Choice of Data:** The sandbox will allow users to select one of three options: input their own dataset, select one of the benchmark datasets in fair ML (e.g. Adult Income (Kohavi & Becker, 2017)), or synthetically generate a dataset. For custom data, the user will be asked to indicate which columns are to be used as group, outcome and feature columns. For synthetic data, which is the recommended option at this time, we provide a rigorous helper file which allows users to customize how the dataset is built. For example, we permit users to determine the number and quality (e.g. categorical or numeric) of features, the distribution of values for each feature, and the proportion of examples in different protected groups. The protected attribute is assumed to be binary. In addition, users can choose how labels are generated where label distributions are allowed to vary across groups. If desired, data can be sampled from a causal graph as demonstrated in Appendix C.

2. **Bias Injection:** The crux of our sandbox pipeline is the injection of different types of biases. In this iteration of the tool, we provide the options to inject representation bias, measurement bias, sampling bias, and label bias (Mehrabi et al., 2021; Frénay & Verleysen, 2013) which spans a large portion of the bias types discussed in the fair ML literature. Support for other types of bias will be added in the near future. In addition to injecting bias into the whole data set, the sandbox tool allows for application of biases at the intersection of protected attributes and other variables. For example, users can decide to under-sample only the positively-labeled examples from a group. Users are able to inject multiple biases at once which allows for realistic bias patterns that are multi-faceted in nature.

3. **Model Class Selection:** The proposed sandbox tool is compatible with any machine learning model in the scikit-learn paradigm. We encourage the use of the so-called white-box classifiers, as they allow for greater ease when reasoning about the results obtained throughout the sandbox pipeline and present use cases of the sandbox with logistic regression in Sections 3 and 4.

4. **Fairness Intervention:** We make use of four fairness enhancing algorithms from the Fairlearn package (Bird et al., 2020) covering pre-processing, in-processing and post-processing techniques. First, the `CorrelationRemover` pre-processing algorithm filters out correlation between the sensitive feature and other features in the data. Next, the `ExponentiatedGradient` and `GridSearch` in-processing algorithms operate on a model and are based on Agarwal et al. (2018). Finally, the `ThresholdOptimizer` post-processing algorithm adjusts a classifier's predictions to satisfy a specific fairness constraint.

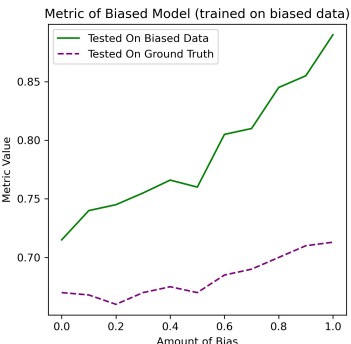

Figure 2: Exemplary visualization generated by the sandbox. We compare the performance of a biased model on ground truth and biased data.

Possible fairness metrics for in- and post-processing algorithms are Equalized Odds, Equality of Opportunity, Demographic Parity, Error Rate Parity, and False Positive Rate Parity. For example, in Section 3, we utilize the `GridSearch` algorithm subject to an Equalized Odds constraint. In Appendix C we consider Equalized Odds, Equality of Opportunity and Demographic Parity metrics.

5. **Evaluation Metrics:** Any scikit-learn supported machine learning performance metric for classification can be utilized in our sandbox framework. Examples include precision, accuracy, recall, F1 score, etc. Additionally, the sandbox also supports fairness metrics for evaluation, such as Equalized Odds or Demographic Parity disparities. For example, we obtain Equalized Odds disparities for the demonstrations provided in Sections 3 and 4.

6. **Visualization:** The sandbox tool outputs several figures including a visualization of the effectiveness of a fairness intervention at dealing with a particular type of bias. We note that various notions of performance are supported including more traditional measures of performance such as accuracy. Figure 2 provides an example visualization output of the sandbox. The figure displays the performance of a learned model in the selected metric (here, accuracy) over different degrees of bias (here, under-sampling examples from one group). Our sandbox allows us to compare performance in two dimensions: (1) Between models with and without a fairness intervention, and (2) On biased data versus unbiased ground truth data. In the figure, we show the latter comparison. We inject under-representation bias into the training data and utilize Fairlearn's `CorrelationRemover` pre-processing algorithm to modify the data by removing correlations between the sensitive feature and the other features before training the model. What we observe is that, if we only evaluate on biased data, then we might be lulled into a false sense of progress and claim that the intervention is improving our model for increasing amounts of bias. However, when we examine the model's performance on the unbiased ground truth data, we see that performance does not improve significantly.

Overall, the sandbox tool regards the initial data set as unbiased and splits into training and test examples. While the training data are injected with bias, data reserved for testing remains untouched. After model fitting and fairness intervention, evaluation metrics and visualizations are provided on both the biased training data and the ground truth test data. The entire process is repeated for different levels of injected bias and, if indicated by the user, for several repetitions in order to obtain reliable average results.

## 3  Case study: can fairness constraints improve accuracy?

A main objective of the proposed sandbox tool is to aid empirical evaluation of the performance of fairness intervention under different biases. There are various special cases in which the effect of imposing fairness constraints has been characterized from a theoretical perspective (Khani & Liang, 2021; Zhou et al., 2021; Du & Wu, 2021). However, results like these usually focus on an infinite data setting and require a vast

array of assumptions which can call their practical usefulness into question. In the coming sections, we use our sandbox tool to empirically replicate a known result from Blum & Stangl (2020) (Section 3) and explore performance beyond the assumptions required for the theory (Section 4). On a high level, we find that an often prohibitive amount of data is required to approximate the infinite data level result. In addition, our exploration suggests that the theoretical result breaks down completely if some of the structural assumptions on the problem setup and bias type are relaxed. The case study demonstrates how our sandbox tool can facilitate understanding of empirical implications of theoretical results and give the user a better sense of what performance to expect in their specific setting.[1]

We note that the case study and explorations discussed in the main text make various simplifying assumptions including an absence of confounding. Appendix C presents supplementary experiments with confounding bias in a more realistic data setting.

### 3.1 Under-representation bias under Equalized Odds constraints

Fairness intervention into machine learning systems is often cast in terms of a fairness-accuracy trade-off. Yet learning from biased data can actually lead to sub-optimal accuracy once evaluated with regards to the unbiased data distribution. Blum & Stangl (2020) theoretically describe settings in which fairness-constrained optimization on biased data recovers the Bayes optimal classifier on the true data distribution. In this case study, we specifically zoom into one of the findings of the paper, that is, Equalized Odds constrained empirical risk minimization on data with under-representation bias can recover the Bayes optimal classifier on the unbiased data. This result requires several structural assumptions on the data generating process as outlined below. We will draw on the described data generating procedure when simulating data for the sandbox demonstration.

**Data generating process**. Let $G \in \{A, B\}$ specify membership in demographic groups $A, B$ where $B$ is the minority group and let $\mathbf{x} \in \mathcal{X}$ be a feature vector from some feature space. We assume there is a coordinate in $\mathbf{x}$ which corresponds to the group membership and write $\mathbf{x} \in A$ if individual $\mathbf{x}$ belongs to group $A$. The respective features distributions are denoted by $\mathcal{D}_A$ and $\mathcal{D}_B$. In order to generate data, we start with a pair of Bayes optimal classifiers $h^* = (h_A^*, h_B^*) \in \mathcal{H} \times \mathcal{H}$ where $\mathcal{H} = \{h : \mathcal{X} \to \{0, 1\}\}$ is a hypothesis class. For a given constant $r \in (0, 0.5]$, we then draw data points $x$ such that with probability $1 - r$ it holds $\mathbf{x} \sim \mathcal{D}_A$ and with probability $r$ it holds $\mathbf{x} \sim \mathcal{D}_B$. Dependent on the class membership, the true label is generated by first, using $h_A^*$ or $h_B^*$ and second, independently flipping the output with probability $\eta < 0.5$. The second step controls the errors of $h^*$ by ensuring that $h_A^*$ and $h_B^*$ have the same error rate and errors are uniformly distributed.

Starting with a ground truth data set including $m$ observations and label noise $\eta$, under-representation bias is introduced by discarding positive observations from the minority group $B$ with some probability. Specifically, for each pair $(\mathbf{x}, y)$ with $\mathbf{x} \in B$ and $y = 1$, the data point is independently excluded from the data set with probability $1 - \beta$. Note that $(1 - \beta)$ is the amount of under-representation bias.

**Recovery of the Bayes optimal classifier**. We first note that recovery of a classifier only pertains to the binary predictions. A Bayes optimal classifier learned from the noisy unbiased data does not necessarily have the same class probability predictions as $h^*$ even in the infinite data setting. To see this, consider the case in which $P(h^*(\mathbf{x}) = 1|\mathbf{x} \in A) = P(h^*(\mathbf{x}) = 1|\mathbf{x} \in B) = 1$ and $\eta = 0.2$. Then, fitting a sufficiently complex threshold based classifier on enough noisy data will result in a predictor $\hat{h}$ with $P(\hat{h}(\mathbf{x}) = 1|\mathbf{x} \in A) = P(\hat{h}(\mathbf{x}) = 1|\mathbf{x} \in B) = 0.8$. While class probabilities differ, both $h^*$ and $\hat{h}$ are Bayes optimal and, in this case, reflect the same binary predictor when selecting a threshold smaller or equal to 0.8.

**Main recovery result**. The derivations in Blum & Stangl (2020) are concerned with fairness constrained empirical risk minimization where an estimator $\hat{Y}$ is deemed fair if $\hat{Y} \perp G|Y = y$ for $y = 1$ (equality of opportunity) or $y \in \{0, 1\}$ (Equalized Odds). Here, $G$ denotes the protected group attribute. In our binary

---

[1]The code generating the results in this Section can be found in the following repository: https://anonymous.4open.science/r/bias-stress-test-sandbox

prediction setting, the Equalized Odds ([Hardt et al., 2016](#)) constraint is equivalent to

$$P\left(\hat{Y}\middle|\mathbf{x} \in A, Y = y\right) = P\left(\hat{Y}\middle|\mathbf{x} \in B, Y = y\right),$$

for $y \in \{0, 1\}$. The main result presented here is based on Theorem 4.1 in [Blum & Stangl (2020)](#) where a proof can be found. We note that this is a population level or 'with enough data' type of result.

**Theorem 1 ([Blum & Stangl (2020)](#))** *Let true labels be generated by the described data generating process and corrupted with under-representation bias. Assume that*

1. *both groups have the same base rates, i.e. $p = P(h_A^*(\mathbf{x}) = 1|\mathbf{x} \in A) = P(h_B^*(\mathbf{x}) = 1|\mathbf{x} \in B)$, and*

2. *label noise $\eta \in [0, 0.5)$ and bias parameter $\beta \in (0, 1]$ are such that*

$$(1 - r)(1 - 2\eta) + r(1 - \eta)\beta > 0.$$

*Then, $h^* = (h_A^*, h_B^*)$ is among the classifiers with lowest error on the biased data that satisfies Equalized Odds.*

### 3.2 Empirical replication using the sandbox toolkit

**Contribution of the sandbox**. The finding in Theorem 1 implies that fairness intervention can improve accuracy in some settings which goes against the common framing of fairness and accuracy as a trade-off. However, Theorem 1 is a purely theoretical result which can make it difficult to assess its usefulness in any specific application setting. For example, the Theorem operates at the population level suppressing issues of sample complexity. In practice it is unclear how much data would be needed for a satisfactory performance even if all the assumptions were met. Our proposed sandbox tool can bridge this gap between theory and practice by providing a controlled environment to test the effectiveness of fairness interventions in different settings. In the case of Theorem 1, the fairness sandbox can help to (1) Give a sense of how fast the result kicks in with a finite sample, (2) Assess effectiveness in a specific data generation and hypothesis class setting , and (3) Understand the importance of the different assumptions for the result.

**Implementation with the sandbox**. We describe how the different modules of the sandbox toolkit are used to empirically replicate the findings of Theorem 1.

1. **Choice of Data**: We opt for a synthetic data set generation according to the exact process described in Section 3.1. This leaves room for several input parameters which can be varied by the user. While some of these parameters determine whether the assumptions of Theorem 1 are met, i.e. the relative size of groups and the amount of label noise, the theorem is agnostic to the number of features, the distribution of features, and the Bayes optimal classifiers and their hypothesis class. In order to simplify reasoning about the results, our analysis focuses on a setting with only three features $x_1, x_2, x_3 \sim \mathcal{N}(0, 1)$ and a linear function class for the Bayes optimal classifiers. The illustration can be readily repeated for more features and a different Bayes Optimal classifier. But this simple example suffices to illustrate some of the key limitations of the theory in [Blum & Stangl (2020)](#). More specifically, the group dependent Bayes optimal classifiers $h_A^*, h_B^*$ are thresholded versions of logistic regression functions

$$\log \frac{p}{1 - p} = b_1 x_1 + b_2 x_2 + b_3 x_3 \tag{1}$$

   for group dependent parameter vectors $\mathbf{b} \in \{\mathbf{b_A}^*, \mathbf{b_B}^*\}$. We set the parameters to fixed values $\mathbf{b_A}^* = (-0.7, 0.5, 1.5)^T$ and $\mathbf{b_B}^* = (0.5, -0.2, 0.1)^T$ which leads to different continuous distributions of probabilities between groups but to approximately the same positive rates when thresholded at 0.5 as required for the theoretical setting of the Theorem. Note that to adhere to the theory, we start out with a threshold-based classifier and subsequently add label noise with $\eta$ (instead of the more common way of turning probabilistic predictions into labels, i.e., flipping biased coins for the binary labels).

2. **Bias Injection:** Theorem 1 is concerned with a specific form of inter-sectional under-representation bias which strategically leaves out positive observations from the minority group. The sandbox is set up to inject this type of bias based on a user specified parameter $\beta$ which determines the amount of bias injected. The addition of further types of biases goes beyond the theory presented in Blum & Stangl (2020) and is empirically explored with the sandbox tool in Section 4.

3. **Model Class Selection:** The theoretical result we are looking to replicate operates on a population level and does not constrain the Bayes optimal classifier or learned model to belong to a specific class of functions. However, in practice we need to select a class of models with enough capacity to express both Bayes optimal classifiers $h_A^*$ and $h_B^*$ at once since the fairness constrained empirical risk minimization requires us to train a single model for both groups. To accomplish this, we select a logistic regression function of the form

$$
\begin{aligned}
\log \frac{p}{1-p} &= b_0 + \mathbf{1}(\mathbf{x} \in A)\mathbf{b_A}^T\mathbf{x} + \mathbf{1}(\mathbf{x} \in B)\mathbf{b_B}^T\mathbf{x} \\
&= b_0 + \begin{bmatrix} \mathbf{b_A} \\ \mathbf{b_B} \end{bmatrix}^T \mathbf{x}',
\end{aligned}
\tag{2}
$$

where $\mathbf{b_A}$ corresponds to the parameters used for rows belonging to group $A$, and $\mathbf{b_B}$ denotes the parameters used for $\mathbf{x} \in B$. The indicator functions are absorbed into the data by reformatting the feature vectors $\mathbf{x} \in \mathbb{R}^3$ to feature vectors $\mathbf{x}' \in \mathbb{R}^6$ with $\mathbf{x}'^T = [\mathbf{x}^T, 0, 0, 0]$ for $\mathbf{x} \in A$ and $\mathbf{x}'^T = [0, 0, 0, x^T]$ for $\mathbf{x} \in B$. Note that the additional intercept $b_0$ increases the capacity of the model and can only help our performance here.

4. **Fairness Intervention:** Recall that Blum & Stangl (2020) analyze the setting of fairness constrained empirical risk minimization. We choose Equalized Odds constrained optimization as fairness intervention in order to mimic the theoretical setting of the result we are replicating. The constrained optimization is performed by scikit-learn unpenalized logistic regression with Equalized Odds enforcement provided by Fairlearn's Grid Search function which is based on Agarwal et al. (2018). For the sake of comparison, we also fit the model from Equation 2 without fairness intervention.

Since in-processing fairness intervention is not always desirable or possible, e.g. sometimes we only have access to biased black-box predictions, we conduct the same experiments with Fairlearn's post-processing method which enforces Equalized Odds by optimizing group-specific thresholds (Hardt et al., 2016). The respective results are discussed in detail in Appendices B.0.1 and B.0.2.

5. **Evaluation Metrics:** There are several relevant evaluation metrics for the case study, all of which are supported by our sandbox toolkit. First, we are interested in the overall and group-wise accuracy of the learned model which is provided for the models learned with and without fairness invention. Second, we evaluate the Equalized Odds disparity of the models in order to demonstrate the effectiveness of the intervention. Following Agarwal et al. (2018), the extent to which a classifier $\hat{f}$ violates Equalized Odds is computed by

$$
\text{disp}(\hat{f}) = \max_{g,y}|\mathbb{E}[\hat{f}(\mathbf{x})|G = g, Y = y] - \mathbb{E}[\hat{f}(\mathbf{x})|Y = y]|,
$$

where $G$ is the protected group attribute. This definition is adapted to a finite data version by inserting the respective sample means for the expected values. Lastly, we want to demonstrate the explicit finding of Theorem 1 which is concerned with the recovery of Bayes optimal classifier. To this end, we compute the fidelity between the predictions of the learned models and the Bayes optimal classifier. The fidelity between two binary classifiers $\hat{f}_1$ and $\hat{f}_2$ with respect to a data set $D$ is defined as

$$
\text{fid}_D(\hat{f}_1, \hat{f}_2) = \frac{1}{|D|} \sum_{x \in D} \left| \hat{f}_1(\mathbf{x}) - \hat{f}_2(\mathbf{x}) \right|,
$$

i.e. as the fraction of examples on which the predictions of the classifiers coincide. The evaluation metrics are output each for the training and test sets. While fidelity results are discussed in detail in the main text, we refer to Appendix A for a summary of accuracy and disparity results.

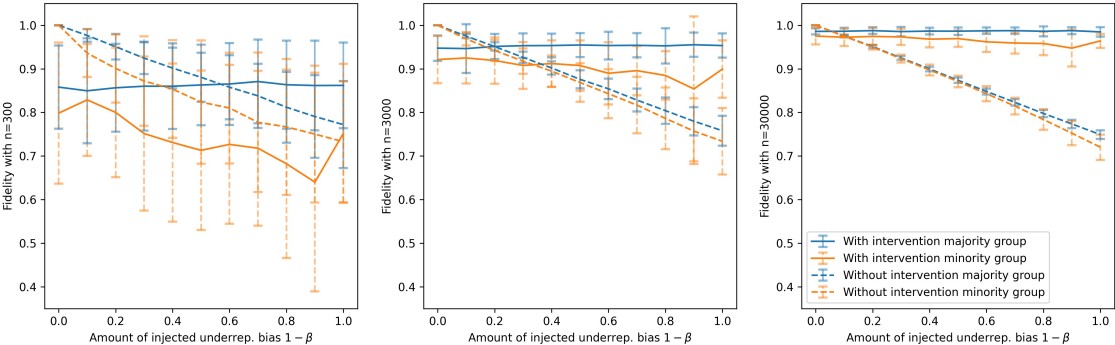

Figure 3: Test set fidelity between Bayes optimal classifier and models trained on biased data with and without fairness intervention using $n = 300, 3000, 30000$ (left to right) samples for training and testing each. Results are reported averaged over 50 simulation runs with error bars for one standard deviation in each direction. We see that Equalized Odds constrained optimization retrieves the Bayes optimal classifier almost perfectly at all levels of bias when using large amounts of data ($n = 30000$) but deviates from the Bayes optimal predictions when trained on $n \in \{300, 3000\}$ data points. The model class used in this example is logistic regression in 7 parameters.

6. **Visualization:** The sandbox tool provides visualizations of the effectiveness of the fairness intervention. In the context of the case study, this consists of figures displaying the accuracies and fidelities to the Bayes optimal classifier of the models learned with and without fairness intervention at different levels of injected inter-sectional under-representation bias.

## 3.3 Empirical results

**Parameter inputs**. The sandbox tool with the described configurations is used to examine the empirical performance of the theoretical result from Blum & Stangl (2020) presented in Theorem 1. In this setting of the sandbox, the user can input several numerical values corresponding to the size of the minority group $r \in (0, 0.5]$, the number of synthetic data points to be generated $n \in \mathbb{N}$, the amount of overall label noise to be injected $\eta \in [0, 0.5)$ and the number of times the whole simulation should be repeated $R$. In each run of the simulation, new data are sampled and injected with bias before the respective models are fit. The whole simulation pipeline is performed based on the input values and performance metrics and visualizations are output to the user.

For the sake of demonstration, we chose $r = 0.2$, $\eta = 0.4$, $R = 50$, which provides one of many examples within the bounds of the theory. In an effort to explore how much data are actually required to obtain the performance promised by the population level theory in our example, we vary the number samples $n \in \{600, 6000, 60000\}$. We note that half of the synthetically generated data are used for model training and half for evaluation and visualization.

**Results**. Figure 3 displays the fidelity results of the sandbox simulation case study measured on the portion of the data sets withheld for testing. We note that fidelity here corresponds to the fraction of test examples that receive the same predictions from the model trained on biased data and the Bayes optimal classifier fit to the unbiased data. No bias is injected in the data used for testing.

We intuitively expect the model fit on biased data without fairness intervention to deviate from the Bayes optimal model especially when large amounts of bias are injected. This is confirmed by the downward slopes of the dashed curves in Figure 3. Theorem 1 implies that fitting the same model on biased data with Equalized Odds fairness intervention recovers the Bayes optimal classifier on the true data distribution. To see that the assumptions of the Theorem are met in our example, note that we selected the Bayes optimal classifiers $h_A^*$ and $h_B^*$ specifically to have equal base rates (see Equation 1), and that our choice of parameters $r = 0.2$ and $\eta = 0.4$ fulfills $(1 - r)(1 - 2\eta) + r(1 - \eta)\beta > 0$ for all levels of injected bias $1 - \beta \in [0, 1)$. We

would thus expect the fidelity of the models with fairness intervention to be 1 for all levels of $1 - \beta$ which is only partially supported by Figure 3. For small amounts of training data ($n = 300$), the average fidelity over simulation runs and levels of injected bias only reaches a level of 0.837 with even poorer performance in the minority group. In cases with 90% of positive minority examples deleted from the training data, the model learned with fairness intervention on average only classifies about 64% of the minority test examples the same way as the Bayes optimal classifier. In addition, results vary significantly over simulation runs leading to many instances with little to moderate amounts of injected bias in which the model learned from biased data without intervention is closer to the Bayes optimal than the model with intervention. With more training data ($n = 3000$), the test fidelity performance of the intervention model increases to 0.942 on average. Yet even in this setting, the biased model outperforms the intervention model if only 20% or less of positive minority examples are deleted from the test data. Only when increasing the training data size to ($n = 30000$), the fidelity of the intervention model reaches 0.982 which is much closer to the results implied by the theory. In this case, the model with intervention outperforms the model without intervention for almost all positive bias levels.

Overall, the findings of the sandbox demonstrate that a considerable amount of data are needed to recover from under-representation bias. We only observed satisfactory results at all positive bias values when 30000 training examples were used for a relatively simple 7 parameter logistic regression model. [2] Many practical applications fall into the range of small data sets and little to moderate under-representation bias in which the intervention model showed to be no more successful in recovering the Bayes optimal model than a model without intervention. The presented case study demonstrates how the sandbox toolkit can help to uncover insights of this type for users who are looking to assess the effectiveness of fairness intervention in their specific application setting.

**Comparison to post-processing intervention**. While Blum & Stangl (2020) specifically call for in-processing intervention, fairness constrained risk minimization is not the only method that targets Equalized Odds across groups. Since post-processing strategies are desirable in some cases, we repeat the same experiments with the threshold based post-processing Equalized Odds algorithm from Hardt et al. (2016). Note that this corresponds to changing the configuration of step '(4) Fairness intervention' in the sandbox pipeline while keeping the fairness metric fixed. The results from this analysis are discussed in Appendix B.0.1 and indicate a very similar performance to the in-processing method.

## 4 Exploration of other forms of bias

Section 3 demonstrates the usefulness of the proposed sandbox tool by empirically evaluating the performance of a theoretical result from Blum & Stangl (2020). For this, we assume the exact setting of the paper with requires a list of structural assumptions on the synthetic data generation, Bayes optimal model and type of injected bias. For example, the replicated finding only considers a specific case of under-representation bias. Real world applications are likely to violate some of the posed assumptions and can carry a number of different biases. In the following, we show how the modularity of the sandbox allows us to explore the performance of fairness intervention beyond the setting posed by the theory. We loosen the assumption of equal base rates in Bayes optimal predictions and inject different types of biases in order to stress-test the efficacy of the intervention. The changes to the sandbox modules discussed in the following refer to the sandbox configuration presented in Section 3.2.

### 4.1 Difference in base rates

**Implementation with the sandbox and parameter values**. The result of Theorem 1 relies on the assumption that base rates are the same across groups which is often violated in practice. We use the sandbox framework to test the extent to which the fidelity of the Equalized Odds intervention is affected by diverging rates and alter the data choice module of the sandbox used in the case study for this purpose.

---

[2]In general, the amount of data required to reliably fit a model increases with complexity of the model class. Many algorithms used in practice exceed the complexity of the model studied here which suggests that even more data are required to observe the desired fairness mitigation effects.

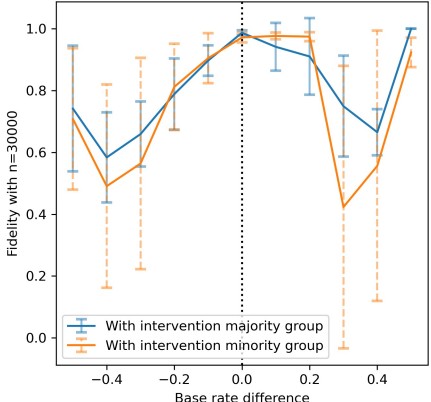

Figure 4: Test set fidelity between Bayes optimal classifier and model trained on biased data with Equalized Odds intervention. Results are reported as an average over 50 simulation runs. Error bars correspond to one standard deviation in each direction. We see that the fidelity between models is generally smaller than 1 if base rates are not the same across groups. In other words, the intervention fails to retrieve the Bayes optimal classifier in these cases.

A collection of data sets with different base rates is generated as follows. We leave the labeling model and effect parameters $b_A^*, b_B^*$ untouched and sample the features $x_1, x_2, x_3$ conditional on group membership with $x_i|(\mathbf{x} \in A) \sim \mathcal{N}(d, 1)$ and $x_i|(\mathbf{x} \in B) \sim \mathcal{N}(0, 1)$ for $i = 1, 2, 3$. Here, $d$ is from a collection of feature mean values selected to lead to evenly spaced base rate differences in $[-0.5, 0.5]$ once the binary Bayes optimal outcomes are computed. Note that the rate of positive outcomes for the minority group $B$ is always 0.5 which justifies the range of the interval.

As in the previous experiments, we set the additional input parameters to $r = 0.2, \eta = 0.4$ and $R = 50$. This aligns with the setting in Section 3 and thus enables us to compare performance across different types of injected bias. That the choices made here are one example among many, they were picked early on to comply with theory and were never changed to obtain specific results. We run the experiment with $n = 60000$ data points at each base rate difference split evenly between training and testing and set the under-representation bias level to $1 - \beta = 0.4$.

**Results**. Figure 4 depicts the test set fidelity of the classifiers trained on biased data with Equalized Odds intervention and the data-driven Bayes optimal model at different levels of base rate difference between groups. The base rate difference is here defined as the base rate of the majority group minus the base rate of the minority group where latter is fixed at 0.5. While the rate of positive Bayes optimal outcomes in the minority group is constant at 0.5, the base rate in the majority group varies between 0 and 1 in our experiment. We see that the intervention model is able to recover the Bayes optimal classifier for a base rate difference of 0 which corresponds exactly to the setting of Theorem 1. The larger the base rate difference becomes in absolute value, the more the predictions of the fair trained model and the Bayes optimal model diverge. The performance in the minority group appears to be particularly poor with a minority base rate of 0.5 and majority base rate of 0.8 leading to minority group fidelity of 0.423 on average. Larger differences in base rates also seem to lead to intervention models with less stable performance which leads to large standard errors.

In order to understand why the result of Theorem 1 does not generalize to settings with different base rates $p_A \neq p_B$, consider that the true positive rate of the Bayes optimal classifier for $G \in \{A, B\}$ on unbiased data takes the form

$$P(h_G^*(\mathbf{x}) = 1|Y = 1, \mathbf{x} \in G) = \frac{(1 - \eta)p_G}{p_G(1 - \eta) + (1 - p_G)\eta},$$

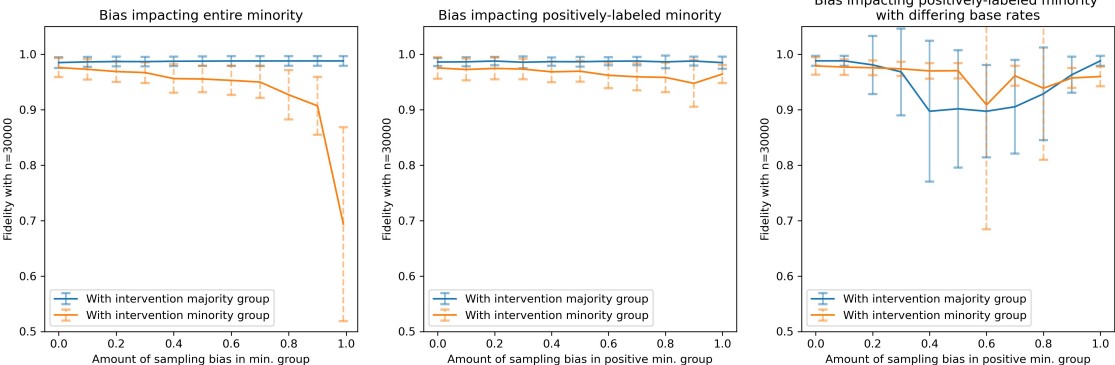

Figure 5: **Sampling bias.** Test set fidelity between Bayes optimal classifier and models trained on biased data with Equalized Odds intervention on 30000 samples for training and testing each. Results are reported averaged over 50 simulation runs with error bars for one standard deviation in each direction. Bias is injected into either the entire minority group (left), or the positively labeled minority group (middle). On the right, bias is injected into the positively labeled minority group and we assume a base rate difference of -0.2

which is different for different base rates $p_A$ and $p_B$. When under-representation bias $1 - \beta \neq 1$ is introduced, the true positive rate for group $B$ becomes

$$P(h_B^*(\mathbf{x}) = 1 | Y = 1, \mathbf{x} \in B) = \frac{(1 - \eta)\beta p_B}{p_B \beta (1 - \eta) + (1 - p_B)\beta \eta},$$

which coincides with the rate for the unbiased data. It follows that the Bayes optimal classifier does not have equal true positive rates, and thus does not satisfy Equalized Odds, on the biased data if base rates are different. It can therefore not be recovered by the fair trained model.

## 4.2 Sampling bias

**Implementation with the sandbox and parameter values**. Our previous discussion of under-representation bias only considered bias specifically injected into the subgroup of examples at the intersection of minority group and positive labels. We extend this setting to under-representation bias in the full minority group, which we will refer to as sampling bias, by altering the bias injection module of the sandbox to remove minority examples with some probability ranging between 0 and 1. Experiments are repeated with equal base rates and with base rate difference of -0.2 which allows us to explore how the performance changes as a difference in base rates is introduced while ensuring that the data still contains examples for both outcomes in each group. We set the parameter inputs to $r = 0.2, \eta = 0.4, R = 50$ and $n = 60000$ to comply with the parameter choices in previous experiments.

**Results**. The results of the experiments for bias injected in the whole minority group, positively labeled minority examples, and positively labeled minority examples with different base rates are depicted in the first column of Figure 5. The left plot shows a decreasing minority test set fidelity with increasing sampling bias in the minority group. With maximally injected bias, 99% of minority examples are deleted and the average minority group fidelity only reaches 0.694. With smaller amounts of bias, the intervention model classifies over 90% of minority test samples like the Bayes optimal classifier. Intuitively, the decreased performance on the minority set can be led back to less available training data for the group. Since we fit only one model for both groups, this leads the predictions for the majority group to be closer to the Bayes optimal predictions than for the minority group. When bias is injected only for positively labeled minority examples, the intervention successfully recovers the base optimal classifier as discussed in Section 3. The right plot of the figure displays the test set fidelity in the case of different Bayes optimal base rates in groups with bias injected only for positively labeled minority group examples. We note that the fidelity here appears much less stable over different runs of the simulation which leads to larger standard errors. In contrast to the

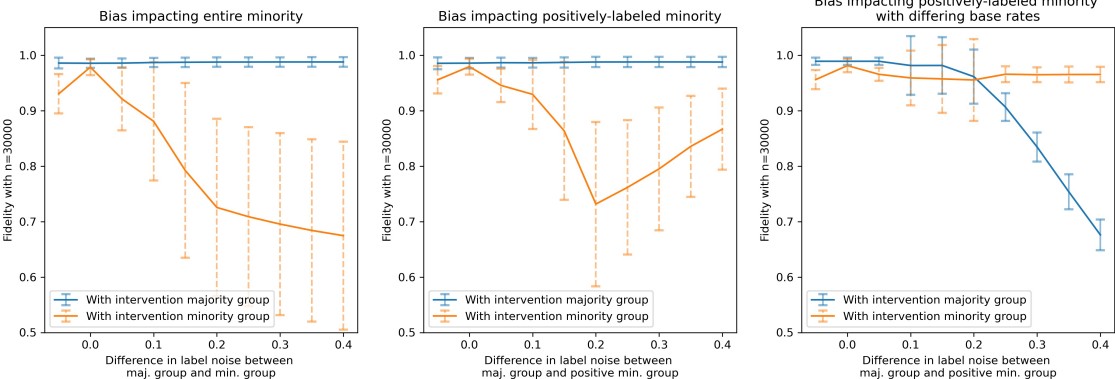

Figure 6: **Label bias.** Test set fidelity between Bayes optimal classifier and models trained on biased data with Equalized Odds intervention on 30000 samples for training and testing each. Results are reported averaged over 50 simulation runs with error bars for one standard deviation in each direction. Bias is injected into either the entire minority group (left), or the positively labeled minority group (middle). On the right, bias is injected into the positively labeled minority group and we assume a base rate difference of -0.2.

setting with equal base rates, the bias injection here impacts also the fidelity of the majority group. Recall that the Bayes optimal classifier does not satisfy Equalized Odds on the biased data in this setting and can thus not be recovered by strictly requiring Equalized Odds. However, the figure suggests a remarkably high fidelity for low amounts of bias in the different base rates case and we hypothesize that the model was faced with large accuracy fairness trade-offs and opted for a small violation of the fairness constraint in favor of accuracy.

### 4.3 Label bias

**Implementation with the sandbox and parameter values**. Recall that our experiments use a noise parameter $\eta$ which represents the probability with which the Bayes optimal label is flipped in our observed labels. So far, this value was chosen independently from group membership. Since data in real-world application often suffers from differential label noise, we test how well the Equalized Odds intervention can recover the Bayes optimal model under label bias. To achieve this, we alter the choice of data module to inject 40% label noise into the majority group to be consistent with the previous experiments. We then change the bias injection module to inject label bias of 0-45% into the minority group. Note that the label bias cannot exceed 50% in order for the Bayes optimal classifiers to be correct which justifies the chosen range. Similarly, we repeat the experiment by injecting constant bias of 40% into both the majority and negatively labeled minority group and vary the amount of bias among the positively labeled minority. In all instances, the test set has 40% label bias throughout like in the previous experiments. The experiment is repeated with different base rates for bias injected into the positively labeled minority examples. As before, we set $r = 0.2, R = 50$ and $n = 60000$.

**Results**. The results of the label noise experiments are depicted in Figure 6. We observe that fidelity performance of the intervention model deteriorates in the minority group as the amount of label noise diverges. This holds true both when bias is injected into the whole group and when bias is injected into positively labeled minority examples. To understand why the intervention cannot retrieve the Bayes optimal classifier, we note that the Bayes optimal classifier does not fulfill Equalized Odds under differential label noise. To see this, assume a setting with $\eta_{\mathrm{maj}} = 0.4$ label noise bias in the majority group and $\eta_{\mathrm{min}} \neq 0.4$ bias in the minority group. The Bayes optimal classifier $h_A^*$ has true positive and true negative rates of 0.6 on the majority group data while $h_B^*$ has true positive and true negative rates of $1 - \eta_{\mathrm{min}} \neq 0.6$ on the minority portion of the biased data. Note that this assumes that base rates are 0.5 like in our experiment, but the same phenomenon with a similar calculation holds true for other cases. If bias is injected into the

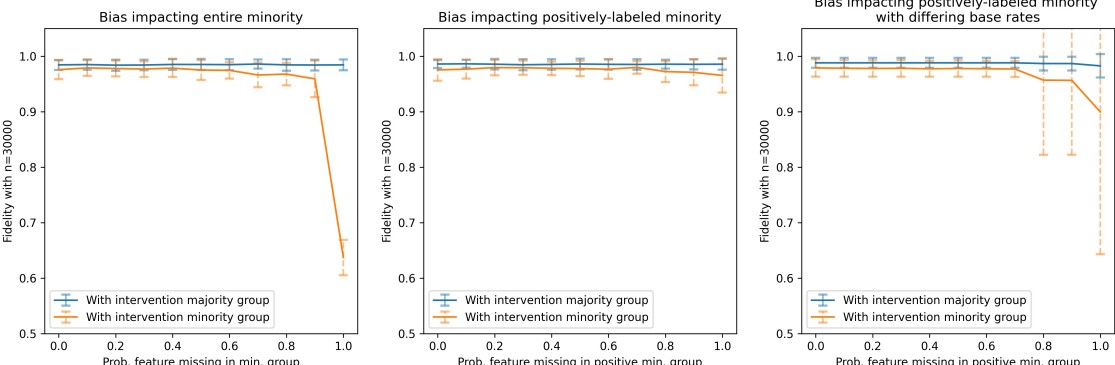

Figure 7: **Feature measurement bias.** Test set fidelity between Bayes optimal classifier and models trained on biased data with Equalized Odds intervention on 30000 samples for training and testing each. Results are reported averaged over 50 simulation runs with error bars for one standard deviation in each direction. Bias is injected into either the entire minority group (left), or the positively labeled minority group (middle). On the right, bias is injected into the positively labeled minority group and we assume a base rate difference of -0.2.

positively labeled minority examples and base rates differ by -0.2, the fidelity curves of both the minority and majority groups are impacted.

### 4.4 Feature measurement bias

**Implementation with the sandbox and parameter values**. Our final exploration focuses on a type of feature noise which is injected in the form of missingness in one of the features. We alter the bias injection module in the sandbox and set feature $x_1$ to 0 with varying probability while omitting the injection of other types of biases. The functionality to enforce different base rates in the data choice module is retained. Feature measurement bias is injected in to the whole minority group or the minority group with positive labels in different variations of the experiment. As before, we choose $r = 0.2, \eta = 0.4, R = 50$ and $n = 60000$. Experiments are repeated with base rate differences of 0 and -0.2.

**Results**. The test set fidelity results of the feature noise experiments are displayed in Figure 7. We see that the intervention model recovers the Bayes optimal model for small amounts of feature missingness while fidelity slightly decreases as more bias is injected. While performance remains above the 0.95 mark for most amounts of injected bias, we observe an average fidelity of only 0.635 if all instances of $x_1$ in the minority group default to 0. Similar to the other types of bias, the intervention model successfully recovers from measurement bias if the bias is only injected into positively labeled examples. In contrast to our observations for other types of biases, a difference in base rates appears to not deteriorate the fidelity by more than 1-2 percentage points for up to 70% feature missingness in a single feature among the minority group examples with positive labels. Assuming our hypothesis from Section 4.2, i.e. with no additional bias the fair learned model on different base rates trades off fairness for accuracy, is true, we conjecture that the performance when injecting measurement bias does not deteriorate quickly because it only introduces very small amounts of Equalized Odds disparity. On a high level, removing the information of one feature leads to a higher concentration around the mean in the predicted conditional probabilities. While this leads to a small violation of Equalized Odds, the fair trained model accepts this unfairness in favor of a high accuracy.

### 4.5 Comparison to post-processing intervention

We repeat the exploration experiments with threshold-based post-processing fairness intervention (Hardt et al., 2016) which corresponds to altering the fairness intervention module of the sandbox tool. The result are discussed in detail in Appendix B.0.2. While the two intervention methods showed to lead to fairly similar results in the original case study setting, this is not necessarily the case when base rates differ or

different types of bias are injected. In those cases, the algorithms face a trade-odd between fairness and accuracy which can lead to different predictions across different intervention methods. For example, we see that the in-processing method yields higher fidelity performance than the post-processing intervention when feature measurement bias is injected as the in-processing method is better in trading off some amount of fairness for accuracy.

## 5 Related Work

**Types of fairness-enhancing algorithms**. At a high level, there are three classes of fairness-enhancing algorithms or fairness interventions: pre, post, and in-processing (Zhong, 2018). These algorithms are applied at different stages in the ML pipeline and can be accommodated by our sandbox toolkit. Pre-processing algorithms modify the data itself and remove the underlying biases which is best suited when training data are accessible and modifiable. Examples of pre-processing algorithms include optimized pre-processing (Calmon et al., 2017), disparate impact remover (Feldman et al., 2015) and reweighing (Kamiran & Calders, 2012). In-processing algorithms operate on the model directly, removing biases during the training process. Examples in this category include the Meta-Fair Classifier (Celis et al., 2019), adversarial debiasing (Zhang et al., 2018) and exponentiated gradient reduction (Agarwal et al., 2018). Post-processing algorithms utilize the predictions and modify the model outputs directly. This approach is best suited when neither the data nor the models are accessible, as it only requires access to black-box predictions. Example algorithms include Equalized Odds post-processing (Hardt et al., 2016) and reject option classification (Kamiran et al., 2012). In this paper, we demonstrate how our sandbox toolkit applies both in-processing and post-processing fairness interventions at the example of a result from (Blum & Stangl, 2020).

**Fairness toolkits**. In order to ease application of fairness interventions in practice, recent work has developed a number of open-source ML fairness software packages or "fairness toolkits" (Bird et al., 2020; Bellamy et al., 2019; Saleiro et al., 2018; Bantilan, 2018; Wexler et al., 2019; Adebayo et al., 2016; Tramer et al., 2017). For example, Fairlearn (Bird et al., 2020) consists of an API to allow researchers and developers to easily use popular fairness interventions (such as Equalized Odds or Demographic Parity) at the three stages of the ML pipeline listed above. Most of these toolkits focus on *fairness interventions*, or how to apply fairness algorithms (Bird et al., 2020; Bellamy et al., 2019; Saleiro et al., 2018; Bantilan, 2018; Wexler et al., 2019; Adebayo et al., 2016; Tramer et al., 2017). A key distinguishing feature of our work is that our toolkit focuses on specific *biases* themselves. Our toolkit allows users to inject biases into their data, uses algorithms from Fairlearn to apply fairness interventions, and then compares to the ground truth. Our toolkit currently uses Fairlearn to apply fairness interventions due to its popularity and ease-of-use. Though, in future development of the toolkit, we plan to add other fairness toolkits, such as AIF360 (Bellamy et al., 2019).

**Algorithms for specific sources of unfairness**. A motivating reason why we focus on injecting specific biases in our toolkit is to evaluate or empirically replicate work or which claims to address specific sources of bias or unfairness. For example, in this paper, we primarily focus on representation bias, measurement bias, and label bias (Frénay & Verleysen, 2013; Mehrabi et al., 2021). See Mehrabi et al. (2021) or Suresh & Guttag (2021) for further detail on more sources of bias or unfairness. To address these sources of unfairness, some have proposed solutions beyond algorithms, such as creating a more representative dataset[3] addressing larger societal inequities. In our toolkit, however, we focus on interventions which can be implemented at the time of training a model, after the dataset has already been created and any broader conditions surrounding model deployment are fixed. Recent work has proposed attempts at algorithmic solutions to remedy specific sources of biases or unfairness. Here, we present examples of the kinds of papers which our toolkit would be able to evaluate. For example, in the context of medical diagnoses, there exists a significant discrepancy in the quality of an evaluation (consider this as the label) between different races (Obermeyer et al., 2019). Khani & Liang (2021) show that removing spurious features (e.g. sensitive attributes) can decrease accuracy due to the inductive bias of overparameterized models. Zhou et al. (2021) finds that oversampling underrepresented groups can not only mitigate algorithmic bias in systems that consistently

---

[3]See, for example, `https://ai.googleblog.com/2018/09/introducing-inclusive-images-competition.html` in response to (Shankar et al., 2017).

predict a favorable outcome for a certain group, but improve overall accuracy by mitigating class imbalance within data that leads to a bias towards majority class. Du & Wu (2021) observes the impact of fairness-aware learning under sample-selection bias. Wang et al. (2021) considers label bias based on differential label noise. Wang et al. (2020) looks at whether fairness criteria can be satisfied when the protected group information is noisy, missing, or unreliable. While our toolkit is able to address many of the claims in these papers, we focus on applying Equalized Odds intervention to data sets injected with the biases listed above in this paper.

# 6 Summary and Future Directions

This work presented the idea and first implementation of a simulation toolkit to investigate the fairness consequences of various forms of biases and identify effective remedies for each the performance of fairness-enhancing algorithms under various forms of counterfactually injected biases. We demonstrated the utility of our tool through a thorough case study of Blum & Stangl (2020). The theoretical contribution of Blum & Stangl (2020) stated that if the source of unfairness is under-representation bias in the training data, constraining ERM with EO can recover the Bayes optimal classifiers on the unbiased data under certain conditions. Our tool allowed us to examine EO constraints under the conditions of Blum & Stangl (2020) as well as a number of new biased settings.

**Lessons from case study**. Through our case study, we established several limitations of the existing theory. In particular, we observed the need for very large volumes of data for the theory to hold. (In our example, we needed 30k training data points for a 7-parameter logistic regression.) Furthermore, our empirical results suggest that the smaller the amount of injected bias, the larger the volume of data needed in order for the fairness-constrained model to outperform the unconstrained one trained on biased data. We emphasize that many practical applications do not satisfy these preconditions (i.e., either the volume of data or the amount of under-representation bias is relatively small). Therefore the theoretical findings of Blum & Stangl (2020), while conceptually interesting, might not be applicable in those practical domains. Another key prerequisite of the theory was the equality of base rates across groups. This assumption is also often violated in practice, and we showed empirically that EO constrained ERM can not recover the Bayes optimal models if base rates differ—even slightly.

**Exploring the implications of various biases and interventions**. We experimented with various forms of biases and assessed the performance of EO constraints in alleviating them. For example, our empirical investigation of *sampling bias* demonstrated how the EO-constrained model struggles to recover comparable performance across groups. We also observed that the constraint could not retrieve the Bayes optimal classifiers under *label bias* either. In terms of the choice of interventions, we contrasted the in-processing method of Agarwal et al. (2018) with the post-processing method proposed by Hardt et al. (2016). The key distinction between these two approaches appeared to be in their ability to trade off accuracy and fairness. In particular, the in-processing method offers a wider range of tradeoff possibilities, while the post-processing method yields fair classifiers but with no error guarantees. When the theoretical conditions of our case study hold, the two methods perform similarly, but they diverge as soon as those conditions are relaxed.

**Scope of applicability and limitations**. First, we should emphasize that the sandbox tool should be understood as an environment to explore the limitations of various fairness interventions in user-specified biased settings rather than a method to obtain fully generalizable results. The insights obtained through this exploration can form the basis of informed hypotheses for further empirical and/or theoretical investigations, but on their own they do not guarantee generalizability. For example, the analysis presented in Section 4 reveals that, at least in our specific experimental setting, EO-constrained optimization cannot recover the Bayes optimal classifier when base rates between groups differ or the data are impacted by label bias. While these findings are not guaranteed to hold in settings beyond the ones studied here, they allow us to surface several limitations of EO constraints as fairness interventions.

Second, we note that the current version of our tool is designed with the intention of helping researchers and students to form a better understanding of sources of unfairness. Our implementation of the data biases mentioned in this work is highly simplified, and it does not capture the complex nature of bias in

real-world data. Addressing bias in specific domains requires prolonged deliberations with domain-experts and stakeholders. Therefore, the results obtained using our tool should not be interpreted in vacuum as the *proof* of efficacy (or lack thereof) for a given algorithmic fairness interventions *in practice*.

**An active-learning module in AI ethics curricula**. In recent years, call for "greater integration of ethics across computer science curriculum" have amplified (see, e.g., Fiesler et al. (2020)). However, instructors without a background in the area may lack the necessary tools to cover these issues in depth (Saltz et al., 2019; Martin, 1997). Our sandbox toolkit can serve these educators as a self-contained and ready-to-use learning module. With the toolkit, students can inject various types of biases into a given dataset, observe the fairness ramifications of the bias, and evaluate the effectiveness of various fairness interventions in alleviating them. By offering a hands-on practice, we hypothesize that the toolkit improves students' understanding of the Machine Learning pipeline, the underlying causes of unfairness, and the scope and limitation of existing algorithmic remedies depending on the type of bias present in the setting at hand. In our future work, we plan to conduct human-subject studies at college-level computer science programs to examine the effect of our sandbox toolkit in achieving FATE-related learning objectives and improving the learning experience.

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
