# OpenReview forum: "A Sandbox Tool to Bias(Stress)-Test Fairness Algorithms"
_TMLR — Rejected by TMLR_

### Review · Reviewer_48wd · 2023-01-24

**Summary Of Contributions:**

This paper presented a sandbox tool to evaluate fairness algorithms. The sandbox consists of several modules. The data module allows the user to import existing datasets or general synthetic data; the bias injection module can inject one or a combination of different types of biases including representation bias, measurement bias, sampling bias, and label bias; the model selection module can import any machine learning models in the scikit-learn paradigm; the fairness intervention module makes use of the fair learning algorithms from the Fairlearn package including pre-processing, in-processing and post-processing techniques; and finally the evaluation and visualization modules provide fairness evaluation metrics and visualization tools. The paper uses a case study to demonstrate how the sandbox can be used to conduct empirical evaluations of theoretical conclusions and the algorithm’s performance when different sources of unfairness are present in the data.

**Audience:**

Yes

**Broader Impact Concerns:**

None.

**Claims And Evidence:**

Yes

**Requested Changes:**

- Add the evaluation of fairness metrics like demographic parity and equality of opportunity in the case study.
- More case studies are preferred.
- Explain the difference between the sandbox and other tools like AIF 360 and What-if Tool.


**Strengths And Weaknesses:**

+ I think the sandbox will be very beneficial to fairness researchers. Currently, there are many theoretical studies on achieving algorithmic fairness, including dealing with the fairness-utility trade-off and impossibility results for different fairness notions. Although those research papers often provide their own simulation studies, it is still beneficial to have a uniform testbed so that we can test the performance of different algorithms under different settings. The key difference between the sandbox tool and existing tools is that the sandbox tool can inject biases into the data and the users can freely configure the types of biases and how they are to be injected.

- Although the sandbox tool seems promising, there could be more experiments or case studies to show its utilities. Currently, there is only one case study in the paper, although different forms of bias are injected. For each form of bias, the authors only evaluate the fidelity of the machine learning model. Other metrics such as different types of fairness metrics are of interest and should be evaluated as well. I am also interested in the trade-off between fidelity and fairness under different types of bias. In addition, more case studies are preferred. For example, the authors can evaluate existing fairness impossibility results to see if different fairness notions can be achieved simultaneously under general and extreme conditions.

- Another issue is that, although the paper briefly mentions the difference between the sandbox and other fair ML tools, it is still clear to what extent the sandbox overlaps with other popular tools like AIF 360 and What-if Tool. I know that bias injection is one major difference. However, for other modules like fairness intervention, evaluation and visualization, it is not clear where the differences are. The authors should also make recommendations on how the users can combine different tools together.

---

### Review · Reviewer_vZjP · 2023-01-30

**Summary Of Contributions:**

This paper targets a very important problem where the conditions for algorithm selection are under-explored in the fair ML field. To close the gap, it provides a sandbox where users can explore the relationship between bias type and fairness intervention. The sandbox contains several modules, e.g., choice of data, bias injection, model selection, fairness intervention, evaluation, and visualization. This sandbox is demonstrated with the case study of Blum & Stangl.

**Audience:**

Yes

**Broader Impact Concerns:**

There are no impact concerns. Actually, I believe this work will promote fair machine-learning research in many domains and stimulate students' interest in fair ml.

**Claims And Evidence:**

Yes

**Requested Changes:**

This framework is expected to be a standalone package with clean and concise documentation such that users without a fair machine learning background are able to play.

It is encouraged to include more types of bias injection and be compatible with mainstream machine learning software, e.g., TensorFlow and PyTorch. I believe it is conceptually feasible as the models are isolated with data and evaluation.

**Strengths And Weaknesses:**

Strengths

1. The proposed framework facilitates bias injection and algorithm testing.
2. It is a practical workflow to evaluate the fairness algorithms in this sandbox.
3. The outputs of this framework are informative for the model developers. The various configurations make this framework flexible to various scenarios.


Weakness

1. It is limited to just exploring the relationships between known bias and existing fairness interventions. It is not clear whether the framework is extensible or how easy it is to be extended.
2. This framework shows its effectiveness in tabular data. It would be more complete to show abilities in image data or sequential data.
3. Usually, the bias evaluation is coupled with performance evaluation. Most existing research assumes the performance is correctly assessed. It is unclear whether the existing work, including this one, is able to detect and evaluate bias if the performance metrics are wrongly selected.

---

### Review · Reviewer_wnXp · 2023-02-05

**Summary Of Contributions:**

This paper provides a simulation tool to stress-test fairness algorithms in a simulated environment. In particular, the simulated environment builds on and extends the setup of Blum & Stangl (2020). The authors provide empirical verification for the claim of Blum & Stangl (2020) that in some cases EO regularization with a proper strength would be the Bayes optimal solution for performance (i.e., there is no tradeoff between fairness violation and performance). They also extend the setup in several ways by adding bias and showing that the Bayes optimality breaks down. The authors release their analysis environment as a general-purpose tool for such stress testing with (1) the simulated environment, and (2) arbitrary datasets.

**Audience:**

Yes

**Broader Impact Concerns:**

I think the authors have done a great job discussing the broader implications of their work throughout the paper, and especially at the concluding remarks!

**Claims And Evidence:**

Yes

**Requested Changes:**

* I think the paper is too long (in its current form) given the technical content that is being put forth. I suggest the authors revise the paper and strive to fit it in ~12 pages (and move some of the details to an appendix), which I hope will make the paper more concise and fleshed out.

* I think *Bias(Stress)-Test* is hard to read. I'd suggest the authors change it to something that is easier to recall.

* Please make connections to the literature on *noisy labels*, *distribution shift*, *adversarial robustness*, *imbalanced demographics* within the context of fairness.

* Please consider expanding the scope of comparisons to reveal insights about how the SOTA in-processing methods generalize. For example, I would be intrigued, for example, if the authors showed that method A achieves a better fairness/accuracy tradeoff compared to method B, however, when corrected for the effect of additional biases, method B indeed gives a better *real* tradeoff.

**Other minor issues**

> (Khani & Liang, 2021) shows that

I'd suggest changing this and other similar usecases to *Khani & Liang (2021) show that*

**Strengths And Weaknesses:**

**Strengths**

* Stress testing the understanding of fairness mitigation and how well it generalizes in the presence of other biases (e.g., noisy labels, unbalanced demographics, and distribution shift) is extremely important, and a tool that could help better understand these problems in practice would be really useful for the research community.

* The proposed tool seems to be very generally usable as discussed in Section 3.

**Weaknesses**

* Unfortunately, the full capacity of the proposed sandbox has not been showcased. The evaluations of the paper cover a tiny fraction of the full generality of the method, as discussed in Section 3.

* Unfortunately, the paper did not really reveal new insights about understanding fairness in the presence of other types of biases. Bulk of the paper is dedicated to empirically replicate the results of Blum & Stangl (2020). While important, I think this would be better fitted for a blogpost, or one usecase out of many in the paper.

* The rest of the analyses on other types of biases introduced in the simulated environment mostly revealed that the Bayes optimality claim of Blum & Stangl (2020) breaks down in such imperfect situations. I think given that for the most part the community is used to seeing tradeoff curves for fairness and performance, this would be considered expected.

* The paper compares in-processing method of Agarwal et al. (2018) with the post-processing method of Hardt et al. (2016), and show that generally the former achieves better tradeoffs between fairness and accuracy. Given that it is considered well-known that in-processing methods give better in-distribution tradeoffs, I would call that finding unsurprising.

* There are several pieces of work that specifically target *noisy labels*, *distribution shift*, *adversarial robustness*, *imbalanced demographics* within the context of fairness. I think at the very least the authors should consider making connections to those pieces of related work.

---

### Review · Reviewer_Nh2q · 2023-02-05

**Summary Of Contributions:**

This paper is proposing a toolbox for assessing the impact of interventions on ML models intended for improving fairness. The tool is intended to have an ability to induce various forms of biases in the data, and then assess whether a particular in-training/post-hoc fairness interventions results in desired results. The data biases focused on i) measurement bias, ii) sampling bias, iii) label bias, iv) representation bias. The toolbox would work with synthetic as well as real-world data, though it appears to have more utility with synthetic data with the ability to introduce custom bias. The authors demonstrate potential utility via case studies, demonstrating practical implications of theoretical results of Blum & Stangl 20. Metrics to assess are primarily disparity and fidelity. Authors further demonstrate case studies with varying base rates, label noise etc.

**Audience:**

Yes

**Broader Impact Concerns:**

I don't have concerns on the ethics of the work, the authors have sufficiently discussed implications.

**Claims And Evidence:**

Yes

**Requested Changes:**

My main suggestion is to restructure the paper a bit. First fully explain the framework in terms of its capabilities.

i) What kinds of synthetic data can be generated (dimensionality, assumptions on the data-generating processes, types of noise etc.)

ii) Go on to explain how the API might allow for i) choice of fairness metrics, ii) choice of fairness interventions, iii) choice of type of method for a particular fairness interventions. For example, I did not realize the toolbox allowed for synthetic data of different base-rates until I read more than half-way through.

iii) Separate out description of capabilities for synthetic and real-world data.

Ideally the above could be the first part of the paper after intro and related work. This is completely separated from specific case studies. Overall *using* the case studies to describe the framework is somehow very challenging on the reader to assess the tool itself.

Then use the case studies purely as demonstration of the capabilities, and introduce necessary theory in the context of the toolbox capabilities.

I believe this would make the paper much more accessible and increase the tool's impact on the community. I hope the authors can incorporate some of these suggestions. I am very happy to revisit my assessment based on the changes.

**Strengths And Weaknesses:**

Strengths:
1. I believe this is a great tool to build and will indeed be useful for practitioners to assess utility of fairness interventions.
2. I believe the results are insightful and I like the choice of the case study that evaluates practical implications of the Theory from Blum & Stangl.
3. Overall presentation is quite good and clear

Weakness:
1. I have a few qualms about the way the paper is structured. I think that makes it quite challenging to assess how extensive the framework is and how widely it could be used.
2. The framework description is not detailed enough. First I would like to know, how general is the kind of synthetic data that I can generate, what features are crucial for such fairness assessments and is the framework's synthetic data generation useful and sufficient for the kinds of potential assessments of fairness interventions I might consider in practice?
3. Then I'd like to know how easy it is to incorporate other fairness notions I might care about, that are not considered through the case studies. Is this sandbox essentially a package that can be extended easily? Or is it primarily an abstract framework. The former is a very tangible takeaway from the paper for the community to build on. On the other hand, the latter is not very general. I believe the goal is the former but I couldn't assess the code because the code link seems to have expired. If you wouldn't mind refreshing the link, I am happy to take another look. The way the paper is presented makes it seem like the goal is in fact the latter. See below on my requested changes, which I feel might enable you to make the presentation more general.
4. Some more egregious types of biases are not accounted for, such as confounding bias. If possible, I would like the authors consider introducing confounding bias (known and unknown) and see the impact of the same interventions (equalized odds and equal opportunity) on the downstream ML models. Here the source of confounding may or may not be the sensitive attribute. If the authors think this is not possible, unnecessary, or too challenging, please provide a justification.
5. On page 7, you say: "While class probabilities differ, both h∗ and hˆ are Bayes optimal and, in this case, reflect the same binary predictor." Is this assuming a threshold of <0.8?

Minors:
There are some typos that could be easily fixed:
1. Page 11: "extend to which fidelity..".
2. Page 3: "as an critical..."
3. I believe you can condense some text non-trivially, there are many places where text is repeated without adding new information.

---

### Decision · Action_Editors · 2023-03-20

**Recommendation:** Reject

**Comment:**

The paper proposes a "sandbox" for thoroughly evaluating fairness algorithms, by allowing for injection of various interventions (e.g., label bias, measurement bias). The authors provide an initial implementation of this idea, and use this to study findings from (Blum and Stangl, 2020), as well as showcase extensions to new settings.

Reviewers generally appreciated the motivation of having a fairness "sandbox"; however, there were concerns about the provided evidence being convincing enough. The "sandbox" comprises two components: the conceptual proposal, and the practical implementation. For the latter, as acknowledged by the authors, the provided implementation is a prototype which misses certain features (e.g., visual interface, seamless integration into existing libraries). The provided code comprises a series of Colabs, which are interesting, but the general reviewer consensus is that these fall somewhat short of being something that would be immediately useful to other researchers with minimal friction. e.g., the individual Colabs appear to have sparse comments, and it would appear to require some effort to modify them systematically to test out a new algorithm.

The former by itself could plausibly be a primary contribution: if it gives researchers in the space some way to systematically study their techniques, and inspire them to implement their own version of a "sandbox", then that could be useful. However, on this front it is not clear if the paper is fully successful. Section 2 offers a nice summary of the different general means of algorithmically intervening. However, without some very concrete recommendations for each --- e.g., specific families of label shift distributions --- it is not clear if these by themselves could be actionably employed by other researchers.

Overall, while the paper is definitely well motivated with an interesting idea, the general consensus is that the current execution falls a little short of being completely convincing.


**Audience:**

The paper is well motivated, and the idea of a "sandbox" could be of interest to researchers in the field.

**Claims And Evidence:**

The "convincing evidence" part is not clear: see Comments.